# Through Thick and Thin: The Meaning of Dementia for the Intimacy of Ageing Couples

**DOI:** 10.3390/healthcare10122559

**Published:** 2022-12-17

**Authors:** Päivi Eskola, Outi Jolanki, Mari Aaltonen

**Affiliations:** 1Gerontology Research Center, Faculty of Sport and Health Sciences, and Open University, University of Jyväskylä, 40014 Jyväskylä, Finland; 2Gerontology Research Center, Faculty of Social Sciences and Centre of Excellence in Research on Ageing and Care, Tampere University, 33520 Tampere, Finland; 3Centre of Excellence in Research on Ageing and Care, Department of Social Sciences and Philosophy, Faculty of Humanities and Social Sciences, University of Jyväskylä, 40014 Jyväskylä, Finland; 4Finnish Institute for Health and Welfare, 00271 Helsinki, Finland

**Keywords:** dementia, intimacy, relationships, couplehood, family care, qualitative research

## Abstract

As the population ages, the number of people with dementia increases. An emerging body of research is focusing on living with dementia and understanding the experience of caring and the care burden. There is much less research on the meaning of dementia from the perspective of an older couple’s spousal relationship and related intimacy. This qualitative study explores the meanings of emotional and physical intimacy and the changes brought by dementia in the couplehood of persons with dementia and their spousal carers. The data comprise semi-structured interviews with 35 persons. The interviews were analysed using inductive qualitative content analysis. Four themes describing the meanings of relational intimacy were identified: intimacy as a striving force, intimacy turning into worrisome behaviour, intimacy as physical and emotional dependency, and intimacy turning into one-sided caring for a partner. Dementia changes the intimate relationship in many ways, but shared affection and long-term partnership help maintain the spousal relationship. While dementia may bring about conflicts and behavioural challenges in an intimate relationship, the couple’s shared intimacy and a sense of responsibility for one another may serve as a resource and support the continuity of couplehood.

## 1. Introduction

The ageing of the population and increase in life expectancy mean that family lives and relationships will change as couples live together longer and grow to a very old age together. Since age is a significant factor in the onset of dementia, the increase in the number of older people is reflected in the increase in the number of people with dementia. Dementia refers to a condition with progressive cognitive impairment, including diagnoses such as Alzheimer’s disease, vascular dementia, frontotemporal dementia, and Lewy body dementia. Different forms of dementia are indistinct, and as a result, mixed forms often coexist [1]. In addition to cognitive and functional decline, dementia affects an individual’s personality, emotional control, and behaviour. Hence, the condition often impacts the everyday interactions and intimacy between partners [2,3,4]. Our study aims to find out how people with dementia and their spouses describe physical and emotional intimacy when the other partner has progressive dementia. We approach the topic from the perspective of both partners, with and without the condition. 

Dementia often brings about dramatic changes to everyday life, as the familiar roles in the couplehood become challenged. Hence, progressing dementia may change everything that was once stable in the relationship [5]. However, the effects of dementia on a couple’s life depend on the type of illness and the symptoms the other partner has, as well as how far the dementia has progressed. Previous research shows that, despite everyday life challenges, both partners may aim to maintain a sense of couplehood [4,5,6,7,8]. In the early stages of dementia, in particular, each member of the couple often strives to continue their own activities as they did before the onset of the condition [8], in addition to their joint activities [6,8,9,10].

As the illness progresses, the spousal roles become challenged, and everyday life responsibilities may bring about feelings of anxiety for both partners when, for example, the gender-related roles that both are used to are crossed [11]. This results in situations where the household chores previously conducted by the other fall to the ‘wrong’ partner, such as cooking, clothing maintenance, or car maintenance [4,5,7,9,11,12]. Moreover, one of the essential elements of the couplehood, namely emotional and intellectual reciprocity, may disappear as the condition progresses [2,3,13,14]. These changes may evoke sadness, frustration, resentment, and guilt as they recall the life and interests previously shared in the couplehood [8,12,15].

One common change in daily life and responsibilities is that the partner without the condition is likely to become a family carer (i.e., an informal carer) at some stage of the illness. This change has consequences for the couplehood [2]. A family carer takes care of a family member or another close relative, who, due to illness, disability, or other special need for care, does not cope with their daily life on their own, outside a professional or formal framework [16]. However, having a partner with dementia does not always mean only negative things. The strong and unshakable sense of togetherness between partners also helps them to face life’s adversities, such as dementia [11]. This continuing feeling of togetherness, despite change, makes the family situation somewhat ambivalent because of the many practical, emotional, and moral pressures involved [17]. Spousal carers may experience negative emotions, such as anger at their spouse with dementia, even though they understand the behaviour is due to the condition [18,19,20].

Dementia may have different effects on the couple’s intimacy. Intimacy is a fundamental part of a human relationship, in which people have mutual feelings of trust, emotional closeness towards each other, and shared experiences that enable a sense of mutual understanding [19,21,22]. Sexual intimacy, especially sexual activities, often decrease due to cognitive decline and related symptoms [2,18,19]. However, some people with dementia may experience verbally, as well as physically, reflected hypersexuality as one of the behavioural and psychological symptoms of dementia, which may be a distressing experience for the spousal carer [23].

Drummond et al. studied female carers and found that for them, sexual identity no longer felt relevant, since physical intimacy was absent from their relationship due to the cognitive or physical decline of their partner [24]. Family carers’ sexual identity may be replaced with a caregiving identity [23,24]. These difficulties related to sexuality caused by dementia are placed in the broader context of well-being, as sexual experiences, identity, and behaviour are essential for well-being and health across the lifespan and part of healthy and successful ageing [25].

Intimacy is often used synonymously with sexuality and sexual activity. However, it can be seen as a broader aspect of the human experience, including emotional closeness [26] and physical intimacy, such as holding hands, hugging, and having fun together [24]. It has been argued that emotional intimacy may even deepen in the early stages of dementia, as exemplified by small gestures of attention by both partners [24]. Based on the results of Sandberg, older couples where one partner has dementia emphasise the importance of a committed, positive relationship, rather than sexual intimacy [27]. On the other hand, emotional intimacy may decrease, due to the stress and fatigue caused by caring that falls on the partner without the condition [18].

The spouse whose partner has dementia may try to find closer physical intimacy and emotional bonds [18] and rebuild the relationship as the disease progresses [4,8,9]. However, dementia is often described as a loss of the person, despite their continued physical presence [3,27,28]. Kaplan calls this the state of being a husbandless wife or wifeless husband [29]. During the later stages of dementia, some spousal carers may look to the future and for new social connections, friendships, and relationships. At the same time, the spousal carer may be confused, due to fuzzy boundaries between marriage and approaching widow- or widowerhood [2,18].

As the descriptions of intimacy above show, intimacy is a multidimensional concept that includes physical, emotional, and cognitive aspects in the relationship. There is still little research on the intimacy of older couples living with dementia [27], especially research that employs the descriptions of people with dementia themselves. Holdsworth and McCabe have called for research on the impact of dementia on relationships, intimacy, and sexuality from the perspective of partners, the person with dementia, and the couple jointly to ensure the development of appropriate information and services that meet the needs of both people in the couple relationship [30]. We aim to add knowledge on these topics with this qualitative study. Our research questions are as follows: what forms does intimacy take in a couplehood in which one partner has dementia, and what changes has dementia brought to their relationship?

## 2. Materials and Methods

### 2.1. Data and Study Design

The data consist of 20 semi-structured, 60- to 90-min-long interviews conducted and audio-recorded in 2019 with home-dwelling persons diagnosed with dementia (n = 15) and their spousal carers (n = 20). The participants lived in different regions around southern and central Finland. Among the persons with dementia, nine were men and six were women. Fourteen spouses were diagnosed with Alzheimer’s disease, two with frontotemporal dementia, and two with vascular and two with mixed dementia. This list includes the diagnoses of the five partners with dementia who did not participate in the interviews themselves. The mean age of those with dementia was 76 years, and that of spousal carers was 75 years. Among the carers, six were men and 14 were women. In this article, we use term ‘couple’ when we refer to both spouses together. The participants were volunteers. In the information leaflet, we asked couples interested in the study to participate. All couples signed for the study represented heterosexual couples with a long history together. All but one couple were married.

Most of the interviews were conducted at the interviewees’ homes. Before the interview, each couple decided whether they would participate in the interview together or separately. Twelve couples were interviewed together and three separately, and five spousal carers were interviewed alone. Three couples felt that they could express themselves more freely if they were interviewed separately. Of those partners with dementia who did not participate in the interview, one husband had recently moved into institutional care. Two of the husbands with dementia were unable to attend the interviews, as their condition had progressed to a severe stage. One of the husbands lived at a different address. In addition, one husband decided not to participate in the research. He was at home during his spousal carer’s interview, but he stayed in another room. Interviews included questions about their current health status and well-being, care histories, life satisfaction, relationship, the role of social networks, social participation, and living conditions. In addition, interviews allowed participants to raise matters that were important to them.

### 2.2. Ethics

The National Ethics Committee of Tampere (Decision 37/2018) approved the study. Participants gave their written informed consent for the interview and its recording. The participants were informed of their right to withdraw at any time. They were also informed of the data collection and handling procedures. All research procedures were conducted according to the General Data Protection Regulation [31] and the Declaration of Helsinki. The names used in data samples are pseudonyms. The state of dementia of the interviewees was assessed as ranging from mild to severe. It was self-reported by the person with dementia and his/her spouse, and some of them also mentioned MMSE scores.

It can be difficult for a person with advanced dementia to understand the meanings and consequences of their own decisions or, for example, the purpose of a research interview. To ensure that persons with dementia who participated in the interviews were aware of what the interviews were about and that they were able to participate in the interviews, data collection was conducted in close collaboration with spousal carers. Interviews were conducted by the two authors who have previously worked with persons with dementia and are, thus, aware of the challenges that dementia can bring to the conversational situation. Both researchers have a long experience working with and interviewing older people. Involving persons with dementia in an interview requires the interviewer to be emotionally sensitive and respectful in the discussion situation [32].

Several qualitative studies of family caregiving touch on care and have addressed the perspective of older family carers [2,33], but persons with dementia have often been excluded from research [34,35]. Our study begins from the idea that it is important to include people with dementia in research to the extent that they can participate, in terms of the stage of the illness [4,8,36,37]. Five of the spouses with dementia did not participate in this research. The reasons for their absence are explained in Section 2.1.

In addition, as Clare et al. argued, it is important in research on older couples’ quality of life to study the quality of the couplehood alongside the individual variables [38]. For people with dementia, participation in research may be difficult because they have difficulties with verbal expression [39]. However, this does not mean that older persons with dementia cannot voice their concerns or present their own views on how the disease affects their lives when given the chance to speak out [40]. People with dementia can express themselves in research if appropriate methods are used [41]. Involving people with dementia as partners in research is good research practice and an important ethical guideline [42].

### 2.3. Analysis

In this article, we focus on the questions of what forms intimacy takes in a couplehood where one partner has dementia and what changes dementia has brought to their relationship. The interviewers addressed the relationship by asking how the interviewees would describe it. However, talk about their relationship came up spontaneously during the interviews. For example, when a partner with dementia was asked how they are, they occasionally started talking about their relationship. Moreover, the question of what brings resources to their lives could be answered by descriptions related to the couplehood.

The data were analysed using inductive qualitative content analysis, which proceeds in three stages: preparation, organising, and reporting [43]. As two of the authors conducted the interviews, the data were very familiar. The analysis process proceeded as follows: Interviews were recorded, transcribed verbatim, read in their entirety several times, and coded using Atlas.ti. The purpose of the preparation phase was to select the unit of analysis and make sense of the data [43]. In our study, the units of analysis were, first, the words I, me, and we, and then the sentences that talked about relationship-related issues. In the organising phase, the text was coded by open coding, according to what was mentioned regarding the relationship and its changes, including humour, sex, and bickering among other subjects. The codes were then grouped, and subthemes were formed according to their content (e.g., sexual interaction, reciprocity, humour, and disruptive behaviour). In inductive content analysis, the researcher must decide through an interpretation which things to put in the same theme or category when formulating the themes. The last phase, abstraction, means formulating a general description of the research topic by generating themes until the main themes are reached [43]. Four main themes were formed from the sub-themes, which described the intimacy between couples or the change that occurred in it. The aim of the analysis was, in particular, to give room to interviewees’ own descriptions of their relationships and the physical and emotional intimacy between them. The descriptions of both spouses were analysed together.

## 3. Results

Intimacy in a couplehood can take many forms related to the individual symptoms and stage of the dementia. From the data, we identified four main themes linked to physical and/or emotional intimacy: intimacy as a striving force, intimacy turning into worrisome behaviour, intimacy as physical and emotional dependency, and intimacy turning into one-sided caring for a partner. Dementia progresses individually, which is why the themes we identify from the data may occur simultaneously or sequentially and are partly overlapping, depending on the stage of the partner’s dementia and the history of the couplehood. Next, we present the themes from the perspectives of both the persons with dementia and their spousal carers.

### 3.1. Intimacy as a Striving Force

Most of our interviewees highlighted their long lives, during which physical and emotional intimacy had become an integral part of their relationship. The interconnection of physical and emotional intimacy meant doing everyday household chores together, doing fun things as a couple, and being in close physical contact in other ways, too, such as napping together, cuddling, and holding hands, all of which show how tightly intertwined emotional and physical intimacy is in the couples’ daily lives. The data showed that continuous physical and emotional intimacy carries the couple through the difficulties caused by dementia. Here, one of the wives, Elina, describes how the couple has fun while walking during daily outings.

Elina (Erkki’s wife, carer, couple interview):*But we have so much fun when we walk in the city hand in hand. Erkki always wants to go there, where the green lights are, and he walks in a zigzag and pulls me behind*. [Laughs].

Physical intimacy also includes sexual intimacy: hugging, kissing, touching each other, sleeping in the crook of their partner’s arm, and making love, as Sirpa and Sauli describe in their account. They also express their intimacy by saying they love each other every morning. These types of descriptions show how emotional, physical, and sexual intimacy are intertwined, overlapping, and, at least partly, build on each other.

Interviewer:
*What is the most important thing in your life that adds to your personal resources?*


Sauli (husband, carer):*It probably is that we say each morning how much we love each other*.

Sirpa (wife):*Yes, yes*.

Sauli:*And then we give each other hugs*…

Sirpa:[Give a laugh] *Yes*.

Sauli:…*throughout the day*.

Sirpa:*Yeah, and when everything is now so intimate. Yes, we have made love almost at least once a week, but now it’s probably been two weeks. I’ve thought that we should soon*…

Sauli:*You have a diary over there*. [Laughs].

Sirpa:*Yes, should we… But really, physical intimacy is terribly important*.

Sauli:*Yeah, yeah. And we have the habit that when we sleep*…

Sirpa:*Yes. We sleep together and*…

Sauli:…*so, just head in the crook of the arm*…

Sirpa:*Yeah. Yes*.

Sauli:…*overnight, until the side goes numb, and you have to switch your position*. [Laughs].

Sirpa:*Yes*.

This couple is a rare example of how both partners emphasised the importance of physical intimacy, including sexual intimacy, in their marriage, which is only accentuated by the wife mentioning that it had been almost two weeks since they last made love. Talking directly about sexual activity was quite rare, and most partners, if they talked about it all, used circular expressions to do so.

However, the analysis showed that dementia causes difficulties in maintaining physical and emotional intimacy, which affects the nature of the relationship in which these forms of intimacy were intertwined. For example, some partners slept in separate rooms because of the restlessness of the spouse with dementia. Because physical and emotional intimacy are intertwined, the absence of physical intimacy complicates emotional intimacy. Helena, wife and carer, described how, as the symptoms of dementia developed, her physical intimacy or hugging did not produce reciprocal physical or emotional intimacy.

Helena (Heikki’s wife, carer, interviewed alone):*Reciprocity is lacking, and the relationship has become one-sided. My partner has become completely dependent on me and my care. I often wonder what is going on in his brain, what he still feels when he is unable to express himself with words, though he can express himself by deeds. He says thank you every night for his meal and that this or that was good. In those words, he says that I have taken care of him and taken care of everything alone. Well, in some way, he gives feedback. But the reciprocity is that I am the one who goes and hugs him. His emotional life towards other people has faded*.

Helena highlights how dementia has diluted her husband’s emotional life, making him unable to respond to his wife’s attempts at mutual physical or emotional intimacy. Therefore, reciprocity becomes complicated, and the responsibility for maintaining intimacy remains with the caring spouse.

### 3.2. Intimacy Turning into Worrisome Behaviour

The data revealed how dementia changes one’s personality and affects behaviour. All the spouses had noticed this, and the wives, in particular, described situations where their husband with dementia could even behave in a disruptive or aggressive manner. We found that sometimes a person with dementia recognises the changes in their own behaviour, such as having become short-tempered. The disruptive behaviour discussed in the interviews could be physical and/or verbal and emotional.

Erkki is one of the husbands with dementia. According to him, a person with dementia wants to be left in peace and easily recognises if there is something wrong with the general mood. He feels annoyed and nervous when his wife gives him advice on daily chores. In their relationship, he has always been more quick-tempered than his wife, and with dementia, this trait has been accentuated in him. Erkki also refers to the area in which he grew up and explains his temper through it.

Erkki (husband):*A person with dementia just likes to be left in peace as activities begin to decline. General atmosphere affects. And it stays in my mind. When there are bad vibes, you suffer from them and get angry easily*.

Elina (wife, carer):*Well, yeah. Nowadays, Erkki gets angry. It also happened before the illness. But it is different now. It happens so suddenly, and he looks really cruel*.

Erkki:…*Ostrobothnian (a region in Finland) fierce*…

Elina:*So true, that the fact is that, of course, I can’t say things the same way you do* [Laughs]*, you say things so it can even hurt me verbally*.

Erkki:*Yes, I may say something sharply*.

Elina:*Yes, that’s how you say something, very sharply. But I have noticed that it’s good that after that you act as if nothing had happened*.

Increased physically disruptive behaviour by the partner with dementia had negatively affected emotional intimacy. Disruptive behaviour was understood as a symptom of dementia, but this understanding did not remove the fear of physical aggression and its effects on the relationship. For most couples, this was a new situation, and it meant a change in their relationship. Maire, wife and carer to Markku, described how she felt frightened when her husband behaved surprisingly aggressively. He had never behaved aggressively towards anyone before he fell ill with dementia.

Maire (Markku’s wife, carer, interviewed alone):*This is not a nice situation, but there is nothing you can do about these diseases. But there will be difficulties. And now, when that fist gets swung in front of my face a little too often, it’s starting to get scary*.

Tyyne, Toivo’s wife and carer, described how she must guide her husband to get dressed in the same way she helped their children decades ago. In Finland, it can be dangerous to wear summer shoes during the winter, but this fact can be forgotten by the spouse with dementia. Therefore, dressing properly for the weather is a factor that can create conflict between partners. The wife advising the husband how to dress sometimes provoked disruptive behaviour.

Tyyne (wife, carer):*We sometimes have a situation that is a bit tense when he often refuses to put his winter shoes on. This morning he did put the proper shoes on, but many times I have had to turn him back from the front door when I find out he’s wearing sneakers, summer sneakers. In that situation, I’m about to get hit when he doesn’t accept that*…

Toivo (husband):*Hmm, I fell*.

Tyyne:*Yes, you did*.

Toivo:*When the shoes were so slippery. There was snow on the ice, ice with a surface like a mirror*.

Toivo, Tyyne’s husband, calmly stated that he fell because his summer shoes were so slippery. He did not comment in any way on his wife’s description of how he behaved aggressively towards her.

One of the wives interviewed alone, Aila, described how, because of her husband’s aggressive verbal behaviour, she had even considered divorce. However, as the dementia progressed, she felt she could no longer leave her husband because the responsibility for the spouse was too great.

Aila (wife, carer, interview alone):*Divorce was considered, if I’m honest. But I think I should have done it a little earlier. But, once this situation has calmed down, there is responsibility for the other person, I’m no longer able to make such decisions at this point*.

Pekka, a husband with dementia, and Pirkko, Pekka’s wife and carer, were interviewed separately at Pirkko’s request. What is noteworthy is that they described their daily lives differently. Pekka said that there were no problems with his behaviour, but Pirkko described how Pekka had started talking in a nasty tone, which is evident in the following excerpts.

Pekka (Pirkko’s husband):*I behave quite calmly*. [Laughs] *No, never, there are no problems with my behaviour. I still behave like I used to*.

Pirkko (Pekka’s wife, carer):*There is nothing wrong with him. I’m the one who forgets. Then he said in a nasty way this morning, there’s probably a third person here in our home. I responded to him that, yes, there is. We have Mr Alzheimer here. He attacked me verbally: “You are the one who doesn’t remember anything!” If the task goes beyond his understanding, for example when we were in the store stacking groceries at the counter, and I asked him to put the groceries stuff over there to make things easier. He flinched: “shut up, bitch”. Hmm. So, we are at that stage now*.

Interviewer:
*Yeah. So, he certainly didn’t behave in this way in the past?*


Pirkko:*No. No. Not at all. This kind of behaviour is new*.

Spouses’ perceptions of disruptive behaviour may differ, or at least they talk about it differently. Pirkko reported that her husband had not behaved verbally aggressively before becoming ill with dementia. With dementia, her husband Pekka gets nervous easily and suspects they have a third person living at home. Jealousy and suspicion are symptoms of dementia. However, later in the interview, Pirkko still described their couplehood as a caring relationship. This is discussed more in the last theme: intimacy as turning into one-sided caring for a partner.

### 3.3. Intimacy as Physical and Emotional Dependency

The data showed how spousal carers are mostly available to their partners around the clock. Depending on the type and severity of the condition, the spousal carer needs to keep an eye on the partner with dementia, or the partner with dementia always wants to be close to the carer. According to the participants, continuous physical closeness is burdensome, and spousal carers wished to have some distance from the situation and occasionally be on their own. Some spousal carers felt so strongly about this situation that they described it as being ‘like imprisonment’ or ‘having a ball and chain´. Burdensome elements consist of the need to be present and available for the needs of the partner with dementia around the clock. In most marriages, spouses have some time for private activities or their own interests, but dementia may change this. Spousal carers have little to no time for themselves, other social contacts, or their own interests.

The partner with dementia might require constant attention, as is the case with Pirkko and her husband. Pirkko, the spousal carer, described how her husband might talk and talk all day, or he is constantly asking something. Persons with dementia commonly repeat a question after they have already received the answer. The data showed that recurrence and repetition were seen as extremely tiresome by the caring spouse. In some cases, the spousal carer gets a moment to themselves if the partner with dementia attends a peer support group or some other event on their own.

Pirkko (wife, carer, separate interview):*When my spouse is in a memory group, I have an hour or two for myself. We have a peer group for carers. Some carers say the spouse does not talk about anything for days. He* [Pekka, Pirkko’s husband] *chatters all the time. He talks and talks. And, if you answer, then the same question will be asked again. It is constant. And I have a break. Just a minute*.

Almost all spousal carers described hoping for even a small amount of time on their own, but none of the partners with dementia said they wanted their own time or space. Some carers described how their situation meant they have no time for themselves at all.

Taina (wife, carer, separate interview):*Before, I felt like I was in jail. Because the time when he had those digestive problems, it was difficult to go anywhere. After all, now I have my own time every day when he spends a lot of time in bed, so I think it’s my own time*.

For Taina, the period of her husband’s stomach problems with dementia had meant being tied to him all the time. Paradoxically, her life has changed positively because her husband has started spending more time in bed, allowing her to be alone. In some other interviews, dementia meant the relationship had started to mean non-stop togetherness and the constant presence of the spouse. The reasons for this were many, as became evident in Aila’s interview.

Aila (wife, carer, interviewed alone):*I have such a rope around my leg. Life has come to the point that I no longer have the freedom to do anything. I’m constantly worried. And this is where jealousy comes in. I wish I could be at home alone for one day. You know how I get annoyed with Olavi* [her husband] *sometimes, so I say, go somewhere, even for a bit, so that I can be alone for a little while. But he is not able to go anywhere. I’d enjoy being alone sometimes. We should do something to make that possible. If there was a nursing home where my husband could be taken for a day or two. That’s all I need*.

It was evident that constant physical closeness and being there for the spouse with dementia all the time burdened the patience and peace of mind of the spousal carers. Sometimes this led to the desire for physical distance, for example, in the form of temporary respite care for the spouse with dementia. Aila would like her husband to be taken to a nursing home for one or two days so she, as a caring spouse, could be home alone without caring responsibilities.

Although their spouse’s dementia had already progressed to a more severe stage, some of the spousal carers were able to continue, for example, a long-term hobby because the partner with dementia managed that time alone at home. Maintaining one’s own time and independence, if only to some extent, can be important for the spousal carer’s well-being. This is evident in Mikko’s account, where he describes how continuing his long-term hobby was important to him. Doing so was made possible by the condition of his wife, which allowed her to be alone at home for a few hours, and Mikko could be confident that she managed being home alone during that time.

Mikko (husband, carer, couple interview):*Well, hobbies (choir), they are like twice a week. There is no time for thinking about household things. Fortunately*.

Singing in the choir is an important hobby for Mikko. He has been involved in choir activities ‘probably for 60 years’, as he stated in their couple interview. Mikko was involved in two different choirs, and the rehearsals lasted three hours at a time. For Mikko, the choir rehearsals were places where he could forget his everyday worries for a moment, since he had to focus on singing. That was his own time.

### 3.4. Intimacy Turning into One-Sided Caring for a Partner

As dementia progresses, the role of the spouse becomes that of a carer who takes care of a patient or, in some cases, similar to that of a parent looking after a child. This theme came up only in those interviews where the spouse´s dementia was more severe. The physical and emotional intimacy of a spousal relationship changes into a different kind of intimacy or closeness that includes caring for the bodily and emotional needs of someone who is vulnerable and in need of help. Carers may perform demanding treatment procedures that not even care professionals can perform without adequate training. Catheterisation is one example of such a treatment, which the spousal carer Valma discussed.

Valma (wife, carer, couple interview):*After all, the fact is that normal home care is not possible because Ville* [her husband] *must be catheterised twice a day, and not every professional in home care can do it. Even in our health care centre, there are not many nurses who can catheterise. I think it’s normal once you’ve learned it and do what the hospital taught you. Ville also says that they* [professionals] *don’t catheterise him properly*.

Valma described how she feels as if she has become a nursing professional and, instead of being a wife, now treats her spouse like a patient. In addition, she thinks she knows how to catheterise better than many health care professionals. Moreover, her husband with dementia is proud of her for doing this demanding treatment procedure.

The spousal carer takes care of the partner’s basic daily needs when it is time. Tuomas, spousal carer, eloquently described the details of his caring duties, while highlighting that he sees no other way forward.

Tuomas (husband, carer, couple interview):*I’m prepared for* [sigh] *what awaits me when everything* [faeces and urine] *goes into a diaper. Well, we have to go to the shower. After all, it’s nothing more than putting her in the shower. I don’t know what will happen, and when I have no other way, I’ll study to become a nurse for a person like her*.

However, Tuomas highlights that he is mentally prepared for the time to come when he will have to take care of all his wife’s basic needs. Different tasks, such as helping with dressing or washing, mean physical closeness, but one which differs from the physical intimacy of a marriage. In addition to the spouse turning into a nurse, the interviewees sometimes described their relationship as a parent–child relation. Thus, sexual and other intimacies related to the spousal relationship disappeared with the dementia. One of the wives, Irja, described how she has lived in celibacy for four years, but at this age, she was already used to it. She described the relationship as ‘*a mother–infant relationship’.*

Interviewer:
*What do you think of your couplehood?*


Irja (wife, carer, interviewed alone):*Well, that’s the mother–infant relationship. Uh, I’ve been celibate for four years now, and at this age it has not been tough. At a younger age, it would have been harder, of course*.

Like Irja, Helena also describes her relationship with her husband as a mother–child relationship that she is in charge of: *‘And I know he’s kind of like a child that I, now, will guide forward.’*

With dementia, the personal emotional intimacy fades. The relationship becomes one-sided when the partner with dementia becomes totally dependent on their wife/husband, and spouses are held together by the caring and responsibility for a spouse with dementia. The data showed that one of the difficulties faced by the spousal carer was emotional loneliness in couplehood, as is exemplified by Aila’s comment in the following extract.

Aila (wife, carer, interviewed alone):*When you asked about this couplehood, well* [sigh]. *I guess that is the kind of caring and responsibility and that kind of stuff, but there’s not much couplehood here when the other is completely helpless. You are not on the same page anymore. And nothing is being discussed*.

In other illnesses, too, the partner may become completely helpless, but as dementia progresses, the personality of the partner also changes, and the spouse becomes a stranger, as Helena described.

Helena (wife, carer, interviewed alone):*I noticed that his personality has changed. That’s what I was about to say, that he’s a completely different person now. He is not a stranger, but quite different. There is this sadness, and you just have to live with it that what used to be there isn’t any more. The emotional side is a different matter. Reciprocity is lacking, and the relationship has become one-sided. The partner has become completely dependent on me and my care*.

The partner with dementia is no longer the same person as they were when the couple married, and the relationship is no longer that of equal partners. The outer surface is familiar, but the inside is foreign. The spousal carer loses a familiar spouse and lives with a stranger who, however, must be cared for.

The data revealed how a partner with dementia and their spousal carer describe the intimacy of their relationship in differing ways.

Pekka (Pirkko’s husband, separate interview):*Well, let’s say that we are such a little team with the wife that we are able to do certain things pretty quickly together and share tasks*.

Interviewer:
*Do you feel that your husband’s dementia has somehow affected your couplehood?*


Pirkko (Pekka’s wife, carer, separate interview):
*Caring relationship.*


According to Pekka, a husband with dementia, they are still a team—a married couple doing things together. According to his wife, Pirkko, this is a caring relationship. That was an example of how, with dementia, the partner did not recognise the illness or the change that had taken place in themselves and in their couplehood, or he did not want to say in the interview that the situation had progressed to this point. In general, with dementia, various changes in behaviour lead to a caring relationship.

## 4. Discussion

Our study expands the knowledge and understanding of how physical and emotional intimacy between older partners manifests and changes with dementia. There are few studies in which older people with dementia have the opportunity to talk about their couplehood [4,8,34,35]. Intimacy is an important part of the relationship, despite dementia. The results of this study combine descriptions of the person with dementia, the carer of their intimacy in a relationship, and the special features that dementia brings to their couplehood. We identified four key themes, which intertwine with each other: intimacy as a striving force, intimacy turning into worrisome behaviour, intimacy as physical and emotional dependency, and intimacy turning into one-sided caring for a partner. The results of this study showed that dementia alters the intimacy of the partnership, but on the other hand, expressions of and needs for intimacy may continue in a similar way as before the diagnosis.

The carer and the partner with dementia provided partly similar and partly different descriptions. The first two themes were raised by both partners, while the latter two were brought forward mainly by spousal carers. Partners with dementia described only a few issues or none related to these themes. It may be that these changes in intimacy are related to a more advanced stage of the disease; thus, the person with dementia does not notice the changes or just does not want to talk about them. However, even if the memory test score is similar, people are always individuals. The causes of dementia vary, and the progression of the condition is individual. In addition, everyone’s current life is built on past life events and the couplehood’s history. The personal qualities of the person with dementia and the life lived together, with all their experiences and feelings, affect how the intimacy of the couplehood takes shape as the dementia progresses. Our data revealed how the severity of dementia affects the maintenance of both physical and emotional intimacy. Physical and emotional intimacy changes shape as a partner’s dementia progresses, but the change does not necessarily progress linearly, and the themes we identified from the data are also partly overlapping.

As has been shown in previous studies [4,5,6,7,8], spousal carers often try to maintain the sense of couplehood, even when the partner with dementia is no longer able to have an equal relationship. Maintaining couplehood involves reciprocal intertwined physical and emotional intimacy, which strengthens the sense of belonging, the idea of ‘us’. In the early stages of dementia, depending on the relationship, emotional and physical intimacy are perceived as very important parts of couplehood. Physical intimacy can be sexual intimacy, but it can also be doing all sorts of things together, including fun and amusing things [27]. Reciprocity strengthens physical and emotional intimacy. However, when a partner with dementia is unable to reciprocally show feelings of intimacy to their spousal carer, intimacy becomes physical closeness, including various treatments or help with basic things, such as getting dressed. Emotional intimacy can turn into a sense of responsibility for the spousal carer because of the long marriage. However, a partner with dementia may not recognise any change in intimacy. For the person with dementia, the couple may still be ‘our team’, see also [8].

The marriages of our interviewees had mainly lasted for decades. They had, at one point, promised to support each other ‘through thick and thin’. In the marriage of some couples, the husband had been dominant, but now he needed help. With dementia, the husband could also become gentler. Three of the wives felt, as carers, that divorce was no longer an option: if they were to divorce, it should have happened earlier, before the onset of dementia. None of the husbands as carers talked about divorce or the possibility of it at any point in the interview. Responsibility and a sense of duty seemed to keep some of the couples together, a result also found in previous studies [4,18,30]. As the memory condition progresses, the person with dementia may experience physical, verbal, or emotional behavioural symptoms (see also 18) that alter both the physical and emotional intimacy between the spouses. As in the previous research of Herron and Rosenberg [20], disruptive behavioural symptoms experienced by the caring spouse emerged in nine interviews, in which the spousal carer was the wife. Disruptive behavioural symptoms were only reported in regard to husbands with dementia. For some reason, husbands as spousal carers did not describe their wife with dementia as behaving in ways that were physically, verbally, or emotionally disruptive. Wives as carers, on the other hand, would talk about their husband’s physically, verbally, and emotionally disruptive behaviour very directly, even in the couple interviews. Three husbands with dementia realised they were behaving disruptively because they talked about it themselves, but they still may not understand the consequences of their behaviour. Sometimes the spousal carers pondered whether their own behaviour and actions caused the disruptive behaviour of the partner with the condition, as Harris et al. also observed in their study [18]. The topic clearly calls for more research on the physical aggression encountered by both female and male carers and how they express it.

The results of our study are in line with those of previous studies showing that spousal carers described themselves as being a living widow/widower, rather than a spouse [2,18], or having a carer–cared-for relationship because of the vast range of social and household responsibilities [2,4,12,17,23,27,28]. However, the partner with dementia did not see their relationship like that. The results show what an emotionally difficult situation the spousal carers may have to live with when trying to maintain the spousal relationship while feeling that, for them, it no longer exists. Some spousal carers even described themselves as mothers. Interestingly, not one carer husband said he was like a father to his wife, although there may have been a protective tone to their descriptions. Instead, husbands as carers might describe themselves as nurses.

Although people with dementia expressed their views less than their spousal carers did, their participation was important, as their views shed light on how they experienced their relationships. They experienced changes in their relationships due to their condition, but the carers expressed more diverse changes. In all, our data revealed that families are different from one another, and their situations vary. It is important to see there is no single uniform formula for how dementia affects couplehood. Marriages are different in their dynamics, couplehood history, and couplehood quality. As dementia progresses, it brings new challenges into the daily lives of the couples. However, a long-shared life history and intimacy can help to maintain couplehood, see also [8]. It would, therefore, be beneficial if couples were also supported in maintaining intimacy in the relationship, despite the challenging symptoms. Despite dementia, spouses are partners with each other, and living together is made up of similar daily chores as before, although these are carried out in slightly different ways than previously and often on the terms of the spouse with dementia.

The relationships of older people, and especially of older people with dementia, have been little studied in family research. As the population ages, the relationships of older people increase in number. In meeting their service and care needs, the partner and quality of the relationship also need to be considered.

## 5. Limitations of the Study

Due to the limited data, the results of our study cannot be generalised to all people with dementia and their spousal carers. However, the experiences of being in a relationship with a person who has dementia are often shared and recognizable. They have common features and similarities. The views of the partners with dementia were less, since some of the spouses with dementia had advanced dementia, so their ability to verbally express themselves was limited. It should also be noted that, since the interviewees had been together for decades, they may have conversational and interactional ways that are related to their relationship, not dementia.

## 6. Conclusions

During long lives together, taking care of the partner’s well-being becomes the key issue, instead of the physical and emotional intimacy usually associated with a couplehood. Behavioural symptoms related to dementia strain both the couplehood and the intimacy between partners. In long relationships, the sense of responsibility for the partner keeps the couple together, even though the relationship is no longer a reciprocal one between two adults, in which both invest and from which both receive benefits. In addition, it could be interpreted that spousal carers act this way because they think it is morally right. However, if the couplehood was previously reciprocal, the sense of belonging between the partners is strengthened, even if dementia has become a third party with unpleasant side effects.

## Data Availability

Data sharing is not applicable. All interviews were conducted in Finnish. Only selected quotes were translated.

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
