# Peer review of "Through Thick and Thin: The Meaning of Dementia for the Intimacy of Ageing Couples"

_healthcare, 2022, doi:10.3390/healthcare10122559_

Round 1

Reviewer 1 Report

The study entitled “Through thick and thin: The meaning of dementia for the intimacy of ageing couples”. The aim of the present study explores the meanings of emotional and physical intimacy and the changes brought by dementia in the couplehood of persons with dementia and their spousal carers. The authors bring up an interesting topic. However, it is mandatory to explain/correct the manuscript in some points.  

Point 1: Please add the limitations of the study to the manuscript.

Point 2: I think the results of this study are meaningful, but I wonder if these qualitative findings can be generalized to the elderly with dementia and their family problems.

Point 3: What is the rationale for choosing the four themes (intimacy as a striving force, intimacy turning into worrisome behavior, intimacy as physical and emotional dependency, and intimacy turning into one-sided caring for a partner)?

Author Response

Author´s Reply to Review Report

Reviewer 1.

Comments and Suggestions for Authors

The study entitled “Through thick and thin: The meaning of dementia for the intimacy of ageing couples”. The aim of the present study explores the meanings of emotional and physical intimacy and the changes brought by dementia in the couplehood of persons with dementia and their spousal carers. The authors bring up an interesting topic. However, it is mandatory to explain/correct the manuscript in some points.  

  • We thank You for Your valuable comments which helped us to clarify the text and our arguments. We have revised the manuscript as requested and hope that the analysis process, the data collection process, and the rationale for choosing these participants is clearer now.

Point 1: Please add the limitations of the study to the manuscript.

  • We have added a study limitations chapter to the manuscript (lines 640–648).

Point 2: I think the results of this study are meaningful, but I wonder if these qualitative findings can be generalized to the elderly with dementia and their family problems.

  • This information has also been added to the chapter limitations of the study: Due to the limited data, the results of our study cannot be generalised to all people with dementia and their spouse carers living in different cultural and social environments. However, the experiences of being in a relationship with a person who has dementia are often shared and recognizable which we have aspired to show with references to previous studies. Relationships and changes brought about by memory illness have common features and similarities and we have tried to give in-depth description of these changes.

Point 3: What is the rationale for choosing the four themes (intimacy as a striving force, intimacy turning into worrisome behavior, intimacy as physical and emotional dependency, and intimacy turning into one-sided caring for a partner)?

  • The themes were formed using inductive content analysis and thus the themes are derived from the data. With extensive extracts we aim to exemplify the themes. We have also aimed to refine and clarify more clearly the analysis process. In the lines 194–212 the coding and analysis process has been described in detailed way.

Reviewer 2 Report

Introduction

Well presented with sufficient background to justify the importance of this topic

Line 16, can give some examples of ‘gender-related roles’ to elaborate more about this challenge ‘the household chores previously conducted by other fall to the ‘wrong’ partner’?

Line 68, can elaborate more how ‘The strong and unshakable sense of togetherness between partners’ being a positive thing in one partner having dementia?

Should define the ‘couples’ clearly. If the couples should be married? Or live as long-term partners? If the couples here mean heterosexual couples or can include homosexual couples?

Material and method

In line 133 ‘some felt that they could express themselves more freely if they were interviewed separately’, why the interview was not taken with all couples being together and separately at two different occasions, so that they had a chance to express themselves freely sometimes.

Line 151, how to assess the state of dementia? Who did that, the interviewers? Any method or any guideline used? Or the assessment was done by medical practitioner?

What is the process to identify the four themes of intimacy to be investigated? It should be explained more in details.

Ethical issue

Some parts should be elaborated in the discussion section instead

Result

It was good to list the qualitative findings through reporting the conversation during interview but it would be better also to summarize in a quantitative way like how many couples having the same kind of comments.

Some parts should be elaborated in the discussion section instead

Discussion

It was good that it pointed out some interesting differences in perception between husbands and wives as carers and raised some topics that can be further investigated.

Yet, I found that some points had been mentioned in result section and felt like a repetition here.

Line 576 ‘some of the wives felt’, it would be better to provide the figure, how many over the interviewed wives carers had this feeling.

Overall, it was an interesting topic and one which is important but neglected.

Author Response

Author´s Reply to Review Report

Reviewer 2.

Comments and Suggestions for Authors

  • We thank You for Your valuable comments which helped us to clarify the text and our arguments. We have revised the manuscript as requested and hope that the analysis process, the data collection process, and the rationale for choosing these participants is clearer now.

Introduction

Well presented with sufficient background to justify the importance of this topic

Line 16, can give some examples of ‘gender-related roles’ to elaborate more about this challenge ‘the household chores previously conducted by other fall to the ‘wrong’ partner’?

  • This has been clarified (lines 57–58): like cooking, clothing maintenance, or car maintenance.

Line 68, can elaborate more how ‘The strong and unshakable sense of togetherness between partners’ being a positive thing in one partner having dementia?

  • In the manuscript we wrote (lines 69–70): “The strong and unshakable sense of togetherness between partners also helps them to face life’s adversities, such as dementia.”

Should define the ‘couples’ clearly. If the couples should be married? Or live as long-term partners? If the couples here mean heterosexual couples or can include homosexual couples?

  • The participants were volunteers. In the information leaflet, we asked couples interested in the study to participate. All couples signed for the study represented heterosexual couples with a long history together. All but one couple were married. (Lines 130–132)

Material and method

In line 133 ‘some felt that they could express themselves more freely if they were interviewed separately’, why the interview was not taken with all couples being together and separately at two different occasions, so that they had a chance to express themselves freely sometimes.

  • The couples decided together whether they would be interviewed separately or together. Usually, they wanted to be together in the interview. The spousal carer was there to support her/his partner with dementia. (Lines 137–138)

Line 151, how to assess the state of dementia? Who did that, the interviewers? Any method or any guideline used? Or the assessment was done by medical practitioner?

  • The state of dementia was self-reported by the person with dementia and his/her spouse and some of them also mentioned MMSE scores. More detailed information on dementia was not needed since the study examined meaning of dementia in a relationship as experienced by spouses. (Lines 157–158)

What is the process to identify the four themes of intimacy to be investigated? It should be explained more in details.

  • We have refined the description of the analysis process to describe in more detailed way how it proceeded. (Lines 194–212.)

Ethical issue

Some parts should be elaborated in the discussion section instead

  • We think that ethical questions also justify the importance of this topic, which is why we would not like to move the text to discussion section. In addition, the ethical questions are based on references, which would only come into discussion as new references.

Result

It was good to list the qualitative findings through reporting the conversation during interview, but it would be better also to summarize in a quantitative way like how many couples having the same kind of comments. Some parts should be elaborated in the discussion section instead.

We have added quantities wherever it was possible. The list of additions below:

  • Line 590: Three of the wives
  • Line 598: in nine interviews in which
  • Line 604: Three husbands with dementia

These quantities were already in the manuscript:

  • Line 225: Most of our interviewees
  • Line 264: This couple is a rare example
  • Line 293: All the spouses
  • Line 559: The first two themes were raised by both partners
  • Line 592: None of the husbands
  • Line 601: husbands as spousal carers did not describe

Discussion

It was good that it pointed out some interesting differences in perception between husbands and wives as carers and raised some topics that can be further investigated.

Yet, I found that some points had been mentioned in result section and felt like a repetition here.

Line 576 ‘some of the wives felt’, it would be better to provide the figure, how many over the interviewed wives carers had this feeling.

  • It is now in line 590: Three of the wives.

Overall, it was an interesting topic and one which is important but neglected.

  • Thank you very much.

Reviewer 3 Report

Dear All,

It was with great interest that I read the article on the meaning of dementia for the intimacy
of ageing couples. It is a difficult aspect of life of seniors that affects each person in
a different and unique way.

While revising the paper the following issues should be addressed:

1) The authors should consider standardising the size of the study group (people with dementia vs. their caregivers). It is significant especially when it comes to the structure of the study and qualitative analysis of the descriptions of both spouses.

2) It would be judicious to indicate what kind of research tool was used to assess the state of dementia of the interviewees.

3) It would be advisable to provide information on qualifications of persons conducting the interviews.

4) The authors of the study might want to reconsider whether it is relevant to reference sources that were published some 30 years ago.

Author Response

Author´s Reply to Review Report

Reviewer 3.

Comments and Suggestions for Authors

  • We thank You for Your valuable comments which helped us to clarify the text and our arguments. We have revised the manuscript as requested and hope that the analysis process, the data collection process, and the rationale for choosing these participants is clearer now.

Dear All,

It was with great interest that I read the article on the meaning of dementia for the intimacy
of ageing couples. It is a difficult aspect of life of seniors that affects each person in
a different and unique way.

While revising the paper the following issues should be addressed:

1) The authors should consider standardising the size of the study group (people with dementia vs. their caregivers). It is significant especially when it comes to the structure of the study and qualitative analysis of the descriptions of both spouses.

  • The participants were volunteers and in the recruitment process we made effort to have couples where both spouses would participate in interviews. However, the final decision whether to participate in the interview was done by the couples and at this stage some spouses decided not to participate. Since we thought it important to include as many participants as possible we chose to also interview those participants' whose spouse did not want to be interviewed.

2) It would be judicious to indicate what kind of research tool was used to assess the state of dementia of the interviewees.

  • The state of dementia was self-reported by the person with dementia and his/her spouse and some of them also mentioned MMSE scores. More detailed information on dementia was not needed since the study examined meaning of dementia in a relationship as experienced by spouses. (Lines 157–158)

3) It would be advisable to provide information on qualifications of persons conducting the interviews.

  • We have added more detailed description in lines 166–67.

4) The authors of the study might want to reconsider whether it is relevant to reference sources that were published some 30 years ago.

  • The 30-year-old references (Moss& Schwebel 1993; Timmerman 1991) were essential for defining the concept of intimacy. In defining the concept, we felt it necessary to use original references.
